

# Deer herbivory reduces web-building spider abundance by simplifying forest vegetation structure

Elizabeth J. Roberson[1,*], Michael J. Chips[2,3], Walter P. Carson[2] and Thomas P. Rooney[1,*]

[1] Department of Biological Sciences, Wright State University, Dayton, OH, United States
[2] Department of Biological Sciences, University of Pittsburgh, Pittsburgh, PA, United States
[3] Pittsburgh Sewer and Water Authority, Pittsburgh, PA, United States
[*] These authors contributed equally to this work.

Corresponding author
Thomas P. Rooney,
thomas.rooney@wright.edu

## ABSTRACT

Indirect ecological effects are a common feature of ecological systems, arising when one species affects interactions among two or more other species. We examined how browsing by white-tailed deer (*Odocoileus virginianus*) indirectly affected the abundance and composition of a web-building spider guild through their effects on the structure of the ground and shrub layers of northern hardwood forests. We examined paired plots consisting of deer-free and control plots in the Allegheny Plateau region Pennsylvania and Northern Highlands region of Wisconsin. We recorded the abundance of seven types of webs, each corresponding to a family of web-building spiders. We quantified vegetation structure and habitat suitability for the spiders by computing a web scaffold availability index (WSAI) at 0.5 m and 1.0 m above the ground. At Northern Highlands sites, we recorded prey availability. Spider webs were twice as abundant in deer-free plots compared to control plots, while WSAI was 7–12 times greater in deerfree plots. Prey availability was lower in deer-free plots. With the exception of funnel web-builders, all spider web types were significantly more abundant in deer-free plots. Both deer exclusion and the geographic region of plots were significant predictors of spider community structure. In closed canopy forests with high browsing pressure, the low density of tree saplings and shrubs provides few locations for web-building spiders to anchor webs. Recruitment of these spiders may become coupled with forest disturbance events that increase tree and shrub recruitment. By modifying habitat structure, deer appear to indirectly modify arthropod food web interactions. As deer populations have increased in eastern North America over the past several decades, the effects of deer on web-building spiders may be widespread.

## INTRODUCTION

Indirect ecological effects due to direct interaction between two species that affect a third species (*Wootton, 1994*) often arise due to the actions of dominant species, keystone species, or ecosystem engineers (*Jones, Lawton & Shachak, 1994*; *Pringle, 2008*). *Menge (1995)* reported that indirect interactions account for ~40% of the change in the abundance

and percent cover of species after experimental manipulations of rocky intertidal food webs. These indirect effects occurred coincident with or shortly after direct effects were observed (*Menge, 1997*). Despite their importance, indirect effects can be difficult to detect, particularly in short-term studies (*Hamilton, 2000*). Moreover, indirect effects can be conflated with direct effects and therefore overlooked entirely (*Wootton, 1994*). Here, we examine the indirect effects of a large mammalian generalist herbivore on the structure of a web-building spider guild.

White-tailed deer (*Odocoileus virginianus*, hereafter deer) in North America have increased in abundance in recent decades (*Crête, 1999*; *Ripple, Rooney & Beschta, 2010*; *Bressette, Beck & Beauchamp, 2012*). In the early 20th century, deer were rare or absent from most of the United States (*Leopold, Sowls & Spencer, 1947*). Now, high deer abundance presents several management problems in much of the United States (*Warren, 1997*; *Côté et al., 2004*) including much of Wisconsin and Pennsylvania. Through selective feeding, deer directly affect forest communities by altering species composition and vegetation structure (*Côté et al., 2004*; *Takatsuki, 2009*; *Habeck & Schultz, 2015*). These direct effects have the potential to indirectly alter the abundance of co-occurring animal species (*Rooney, 2001*; *Rooney & Waller, 2003*; *Sakai et al., 2012*). For example, through resource competition, deer can negatively affect the abundance of small granivorous mammals. *McShea (2000)* observed that in years of low food (acorn) abundance, deer reduced the abundance of two common species of rodent by 50%. Similarly elk (*Cervus elaphus*) reduced vegetation cover, thereby causing a decline in the abundance of woodrats, voles, and two species of mice (*Parsons, Maron & Martin, 2013*). Additionally, deer herbivory can alter resource quality for other herbivores by altering plant species composition, or increasing secondary metabolites of particular species (*Vourc'h et al., 2001*; *Nuttle et al., 2011*). A reduction in vegetation cover and vertical complexity alters habitat for birds and other flying species (*Rooney, 2001*). The removal of deer can lead to increased vertical structure and ground cover (*Rooney, 2009*). In studies where deer are removed, ground and shrub-nesting birds increase in abundance (*McShea & Rappole, 2000*; *Holt, Fuller & Dolman, 2011*).

The indirect effect of deer on arthropods may be strong for species that depend on vegetation for habitat (*Stewart, 2001*) because deer browsing reduces the three dimensional structure of the ground and shrub layers of forest habitats (*Habeck & Schultz, 2015*). Vegetation structure is important for web-building spiders, which use woody and herbaceous surfaces as anchoring points for their webs. These anchoring points can be a limiting resource for web builders (*Rypstra, 1983*; *Uetz, 1991*; *Gómez, Lohmiller & Joern, 2016*). *Miyashita et al. (2004)* examined this relationship in forested regions of Japan. They report that the abundance and richness of web-building spiders increased in areas without deer browsing. They attributed this to an increase in vegetation cover, or more specifically, physical structures for anchoring webs. In a follow-up study, *Takada et al. (2008)* found that web-building spiders were more vulnerable than non-web builders to deer browsing.

In this study, we determined how deer affected assemblages of web-building spiders and their habitat structure. We built on previous work (*Miyashita et al., 2004*; *Takada et al., 2008*) by identifying responses of a broader range of web-building spiders to browsing effects. We examined web-building spider assemblages with and without deer, using a
paired exclosure-control design. We surveyed webs, documented vegetation structure, and inventoried spider prey to determine the degree to which deer alter the abundance and composition of a web-building spider guild, their habitat structure, and their prey abundance.

## MATERIALS AND METHODS

### Field methods

We surveyed ten paired exclosure-control study plots located in the north-central and northeastern United States. Four paired plots were located in the Northern Highlands region of northern Wisconsin in Vilas County (46°9′N, 89°51′W) on a 2,500 ha property owned by Dairymen's Inc (*Rooney, 2009*). This site supported high densities of deer throughout most of the 20th century, greatly altering plant community composition (*Rooney, 2009*; *Begley-Miller et al., 2014*). In 1990, four deer exclosures were constructed in a 5 ha, old-growth hemlock-hardwood stand (predominantly *Tsuga canadensis*, *Acer saccharum*, and *Betula alleghaniensis*). Exclosures are 1.8 m tall, constructed of wire mesh, and range in size from 169 m$^2$ to 720 m$^2$. Each exclosure has an adjacent control plot of the same area, but with ambient browsing pressure. The exclosures are separated from one another by a mean distance of 195 $\pm$ 15 m (*Rooney, 2009*). The remaining six paired plots were located in the Allegheny Plateau region in north-central Pennsylvania in Elk County (41°25′N, 78°50′W). In the early 2000s, the Pennsylvania Game Commission constructed and maintained an array of six deer exclosures in State Gamelands 44 and 28 across a 200 km$^2$ area. This forest is part of the Hemlock-Northern Hardwood Association (*Whitney, 1990*), and is composed of second and third growth forests (predominantly *Acer rubrum*, *Prunus serotina*, and *Acer saccharum*). For a more detailed description of the region, see *Horsley, Stout & DeCalesta (2003)* and *Chips et al. (2015)*. All exclosures were approximately 2.25 m tall, ranged in size from 500 m$^2$ to 900 m$^2$, and had an adjacent control plot in a randomly selected location within 20 m of the edge of each fence.

We surveyed our plots for spider webs, and classified spider webs according to their structure (Fig. 1). Spider families can often be identified by the types of webs they build. We did not always identify the spider that created the web because they were not always present. However, we identified the putative family of spider that created each type of web we tallied (*Bradley, 2013*). We classified webs as: (A) funnel web (Agelenidae), (B) sheet web (Linyphiidae), (C) mesh web (Dictynidae), (D) reduced orb web (Uloboridae) (E) vertical orb web (Aranaeidae), (F) tangle web (Theridiidae), (G) horizontal orb web (Tetragnathidae).

In the Northern Highlands region, we sampled spider webs in each paired exclosure and control plot for five days each month in June, July, and August 2013. Each sampling day, we divided each control and each exclosure plot into a 2 × 2 grid of four equal sections. For each section, we randomly assigned a sampling distance (at least 1 m distance to the next section) and angle (0–90°) using a random number generator. At the random point, we established a cylindrical sampling area with a 0.5 m radius and a 2 m height. We used a spray mist bottle to fill the entire area with water. This increased the visibility of all webs. We identified all spiders to family based on web architecture (Fig. 1).
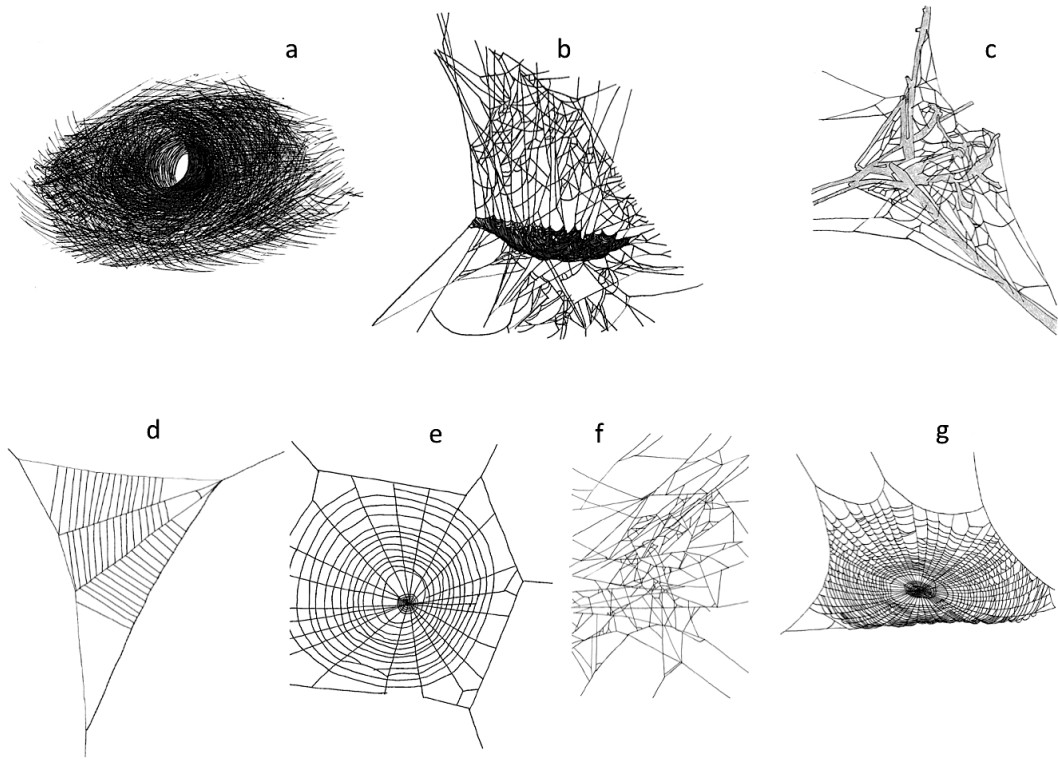

**Figure 1 Web structures (A) funnel web (Agelenidae), (B) sheet web (Linyphiidae), (C) mesh web (Dictynidae), (D) reduced orb web (Uloboridae) (E) vertical orb web (Araneidae), (F) tangle web (Theridiidae), (G) horizontal orb web (Tetragnathidae).** Line drawings by E. J. Roberson.

In the Allegheny Plateau region, we sampled spider webs in each exclosure and control plot once in mid-July and once again mid-August of 2012 using a stratified random approach. We sampled a 20 m × 20 m area within each plot and divided each area into four 10 m × 10 m sections. Within each section, we assigned Cartesian coordinates using a random number generator to determine the sample location, and a new location was generated for each sampling date. At each location, we counted spider webs using the protocol described above. We identified all spiders to family based on web architecture and recorded whether a spider was found in the web.

After we counted and classified webs, we quantified vegetation structure and estimated habitat suitability for web-building spiders using the Web Scaffold Availability Index (WSAI) developed by *Miyashita et al. (2004)*. In natural environments, vegetation provides most of the supports for webs (*Uetz, 1991*). The WSAI quantifies the structural complexity of vegetation. At the center of the same random point used to sample spiders, we horizontally rotated a 1 m stick at 0.5 m and 1 m above the ground. We recorded the number of times each webanchoring structure (branch, twig, leaf, log) touched the stick. The total number of recorded anchoring points at 0.5 m and 1.0 m are recorded as the WSAI 0.5 m and WSAI 1.0 m (*Miyashita et al., 2004*).

At the Northern Highlands site only, we also examined prey availability to web-building spiders using sticky traps. The traps were 23 × 33 cm sheets of clear plastic coated with

Tangle Trap Sticky Coating Aerosol (Tanglefoot Company, Grand Rapids, Michigan). Traps were attached to 1 m high wooden poles. We deployed one sticky trap in each of the established sections within the $2 \times 2$ grid using the same randomization method outlined above. We set up the traps 24 h before the first web sampling day; they were deployed for 5 days each sampling month, after which they were covered with clear plastic wrap and placed on ice. In the lab we counted all insects captured, and measured the body length of each insect specimen to the nearest mm.

## Statistical methods

We examined the effect of deer on habitat suitability for web-building spiders by comparing WSAI values at 0.5 m and 1.0 m above the forest floor inside and outside of exclosures. Because our sampling intensity was greater in the Northern Highlands, we divided the abundance of spider webs by 7.5 to standardize effort (15 days/2 days). We first used two-way nested ANOVAs to determine if studysite location (Northern Highlands or Allegheny Plateau), the deer exclusion treatment, or the 4 subsamples per plot were significant sources of variation in explaining (a) WSAI at 0.5 m and (b) WSAI at 1.0 m. For both WSAI heights, only the deer exclusion treatment was a significant source of variation. We therefore summed values from subsamples into a single value for each plot. We then used an independent two-sample $t$-test to examine differences in WSAI between exclosure and control plots. We conducted two tests, one for WSAI at 0.5 m, and one test for WSAI at 1.0 m. WSAI values were natural log transformed prior to analysis to meet assumptions of normality.

We next examined the relationship between WSAI and spider web abundance using multiple regression. We constructed a preliminary model using WSAI at 0.5 m, and WSAI at 1.0 m as independent variables, and spider web abundance as the dependent variable. We performed stepwise regression with backwards elimination, and used the Bayesian Information Criterion (BIC) to choose the most parsimonious model.

To determine the effects of deer exclusion on the abundance of spiders, we tallied the number of spider webs of each type (Fig. 1) in each exclosure and each paired control plot. We then computed the log response ratio $L$ for the abundance of each web type where $L = \ln(N_{\text{no deer}}/N_{\text{deer}})$. When $N_{\text{no deer}} = N_{\text{deer}}, L = 0$. Negative values of $L$ indicate more spider webs in plots where deer are present, while positive values indicate more webs in plots where deer are excluded. A 95% confidence interval was calculated for each spider web type $L$ to determine if it differed from zero.

We combined values for $L$ from all webs using techniques developed for meta-analysis. Data from each web type were combined to create a mean log response ratio as an effect size, following the procedures outlined in *Hedges, Gurevitch & Curtis (1999)*. To account for among-web type variation in effect sizes, we combined effect sizes from each web type to calculate the mean effect size, or overall effect. The effect size of each spider web type was first weighted by its inverse sampling variance plus a constant, $v_\theta$. The computation of $v_\theta$ is derived from homogeneity analysis and represents variability across population effects (*Hedges, Gurevitch & Curtis, 1999*). To determine if the mean effect size differed from zero,

we constructed 95% confidence intervals. We considered the effects of deer on spider web abundance to be statistically significant if 95% confidence intervals did not include zero.

We counted the number of prey items captured on each sticky trap at our Northern Highlands site, and computed the mean number of prey items per plot. We also measured the length of each prey item to obtain a mean prey size. We calculated log response ratios of prey abundance and size, and computed 95% confidence intervals using the same procedure as above.

We performed an ordination to examine the differences in spider assemblages between exclosures and controls We created a web type × plot matrix with seven web types and 20 control-exclosure plots. Each element of this matrix contained the number of webs tallied. We used PRIMER 6 (*Clarke & Gorley, 2006*) to create a dissimilarity matrix using the Bray–Curtis measure of ecological distance. This non-Euclidian dissimilarity scales from 0 to 100. Web type abundance data were square root transformed prior to the creation of the dissimilarity matrix. This transformation had the effect of down-weighting the most abundant web types, which slightly increases the distance between data points in ordination space for easier interpretation (*Clarke & Warwick, 2001*). This had no measurable effect on subsequent analyses. We used non-metric multidimensional scaling (NMDS) to project the ecological distances among plots, based on 100 random starting configurations and applied a stopping rule when Kruskal stress reached 0.005 (*Clarke & Gorley, 2006*). We used the vegan package in R (*Oksanen et al., 2015*) to further analyze multivariate patterns, based on an untransformed Bray-Curtis dissimilarity matrix of plot composition. We used the adonis function to perform a permutational multivariate analysis of variance (PERMANOVA). This was based on 999 permutations to test for significant differences in spider composition due to deer browsing (exclosure and control plots), and geographic location (Northern Highlands and Allegheny Plateau plots). This was followed with the permutest.betadisper function to test for homogeneity of dispersions among groups. This multivariate dispersion analysis tests whether the average within-group dispersion is equivalent among groups (*Anderson & Walsh, 2013*). This was also based on 999 permutations to test for significant differences in multivariate dispersion due to deer browsing (exclosure and control plots), and geographic location (Northern Highlands and Allegheny Plateau plots).

## RESULTS

We tallied 1,567 spider webs (Table 1). There were about twice as many spider webs in plots without deer compared to plots with deer ($101.9 \pm 10.5$ SE vs. $54.8 \pm 6.0$ SE; $df = 9$; paired $t = 5.16$; $P < 0.001$).

In plots without deer, web-building spiders had more structures upon which to anchor their webs. Web-scaffold availability 0.5 m above the forest floor was over seven times greater in plots without deer ($72.3 \pm 11.9$ SE vs. $10.2 \pm 3.2$ SE), and over twelve times greater 1.0 m above the ground ($39.7 \pm 12.3$ SE vs. $3.2 \pm 1.9$ SE; Fig. 2). Web-scaffold availability did not differ significantly between Northern Highlands and Allegheny Plateau plots at 0.5 m ($df = 18$; $t = 0.38$; $P = 0.70$), or 1.0 m ($df = 18$; $t = 0.97$; $P = 0.34$). The most parsimonious model predicting the abundance of spider webs had a single predictor

**Table 1** Web type, putative family (based on *Bradley, 2013*), number of individual webs identified in exclosure (deer free) and control (browsed) plots.

| Web type | Family | Exclosure | Control |
|---|---|---|---|
| Funnel weavers | Agelenidae | 122 | 136 |
| Sheet weavers | Linyphiidae | 345 | 187 |
| Mesh weavers | Dictynidae | 77 | 30 |
| Reduced orb or line weavers | Uloboridae | 21 | 5 |
| Vertical orb weavers | Araneidae | 133 | 45 |
| Tangle web weavers | Theridiidae | 190 | 97 |
| Horizontal orb weavers | Tetragnathidae | 131 | 48 |

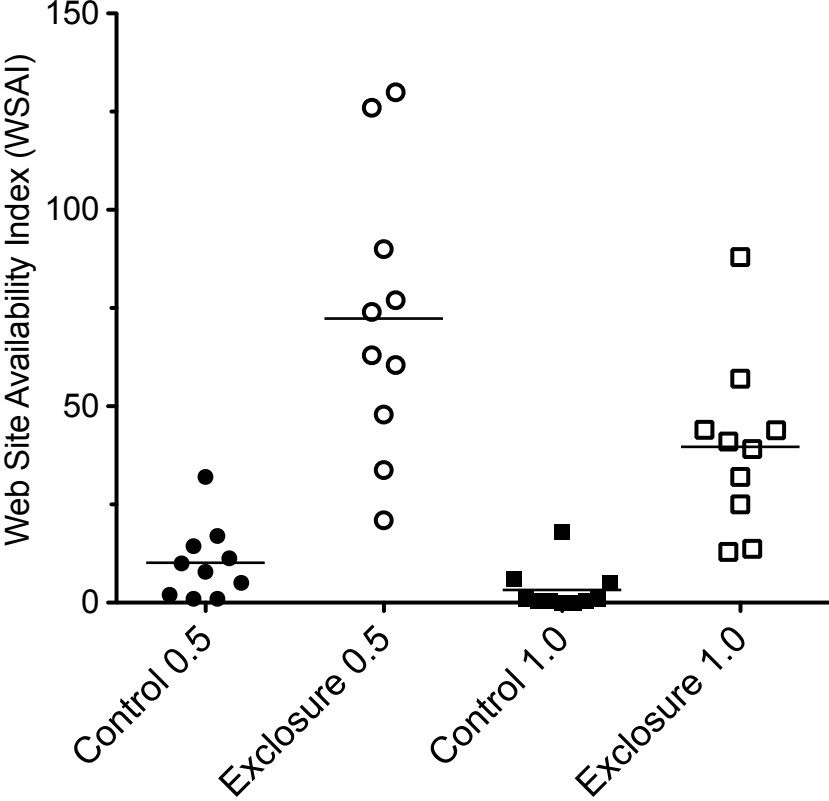

**Figure 2** Web scaffold availability index (WSAI) in control and exclosure plots 0.5 m ($df = 18; t = 5.58; P < 0.001$) and 1.0 m ($df = 18; t = 7.37; P < 0.001$) above the ground. Horizontal lines indicate mean values. WSAI was natural log transformed prior to statistical analysis.

variable: WSAI at 1.0 m (Fig. 3). The WSAI at 0.5 m was not a significant predictor of spider web abundance when the WSAI at 1.0 is taken into account ($n = 20; r^2 = 0.10; P = 0.17$).

In the Northern Highlands region, total prey availability was significantly higher in plots without deer ($886.8 \pm 160.8$ SE) compared to plots with deer ($330.0 \pm 53.8$ SE; effect size $= -0.98 \pm 0.13$; 95%CI $[-1.48--0.65]$). Mean prey size did not differ between plots without deer ($2.80 \pm 0.16$ mm SE) and plots with deer ($2.61 \pm 0.10$ mm SE; effect size $= 0.08 \pm 0.04$; 95%CI $[-0.11-0.25]$).

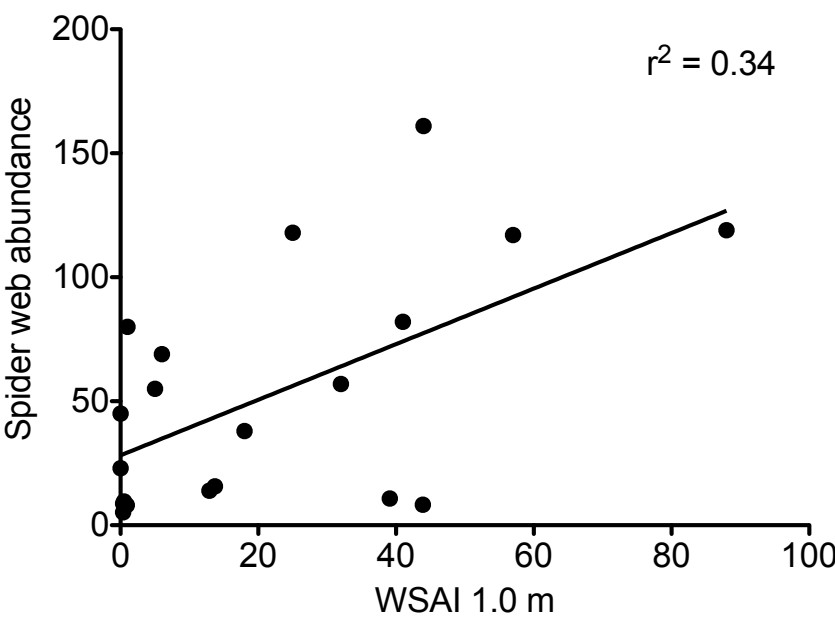

**Figure 3** **The abundance of spider webs as a function of web scaffold availability index (WSAI) at 1.0 m.** Abundance $= 1.12$ WSAI $+ 28.2$ ($df = 1,18$; $F = 9.07$; $r^2 = 0.335$; $P = 0.008$).

Analysis of log response ratios revealed that, with the exception of funnel web-builders, all spider web types were significantly more abundant in plots without deer (Figs. 1 and 4). Vertical and horizontal orb weavers accounted for the largest responses. Both of these groups were nearly three times more abundant in plots without deer. When response ratios were combined for all web types, spider webs were clearly more abundant in deer-free plots (Fig. 4).

A 3-dimensional NMDS solution (Kruskal stress $= 0.07$) provided a better fit than a 2-dimensional solution (Kruskal stress $= 0.15$). While NMDS ordination did not appear to reveal strong differences in the structure of web-building spider guilds between plots with and without deer (Fig. 5), differences were statistically significant (PERMANOVA $R^2 = 0.20$; $P = 0.002$). There were also significant differences in web-building spider guild composition between the Northern Highlands and Allegheny Plateau regions (PERMANOVA $R^2 = 0.13$; $P = 0.014$). Dispersion analysis indicated within-group dispersion did not differ significantly between plots with and without deer ($df = 1, 18$; $F = 0.14$; $P = 0.73$) or between the Northern Highlands and Allegheny Plateau regions ($df = 1,18$; $F = 2.25$; $P = 0.16$).

## DISCUSSION

In both the Allegheny Plateau and Northern Highlands regions, web-building spiders differed in abundance and composition between areas with and without deer. With deer excluded, plots had about seven times the number of anchoring points for webs, and twice as many web-building spiders. Nearly all web structure types increased in abundance when deer were excluded; only funnel weavers were not strongly affected. Prey availability was

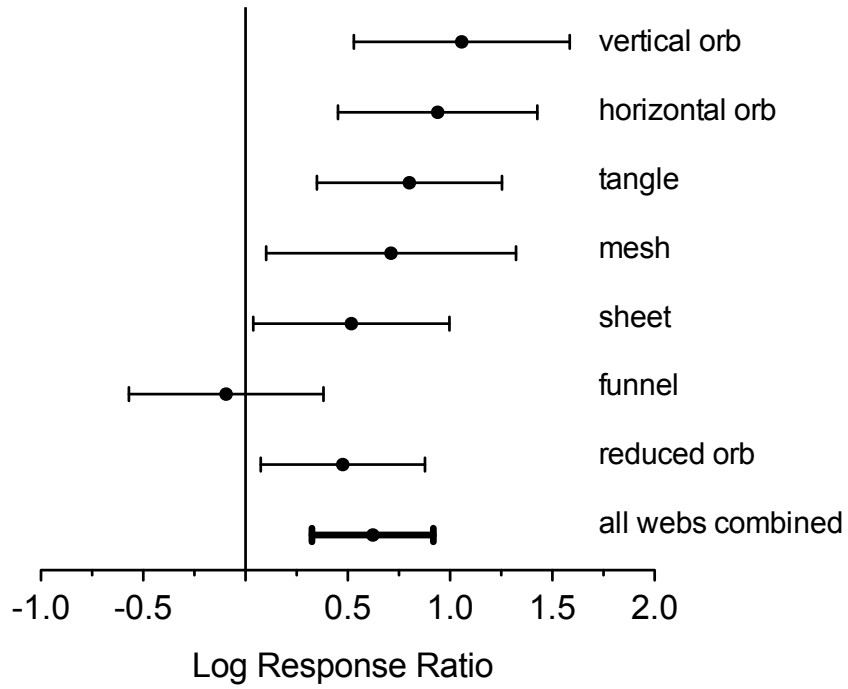

**Figure 4** **Log response ratio (ratios of number of webs in exclosure plots compared to paired control plots) and 95% confidence intervals for all web types.** Positive values indicate greater web density in the absence of deer. Confidence intervals that intercept zero indicate no significant difference ($P > 0.05$).

higher in plots with deer, where the abundance of spiders was much lower. *Miyashita et al. (2004)* did not find a relationship between prey abundance and deer exclusion. This probably reflects a difference in sampling intensity. Our prey sticky traps were 7.5 times larger and deployed nearly four times longer. Our findings suggest that deer, not prey availability, account for differences in web-building spider assemblages between plots with and without deer. When deer are present, web-building spider abundance appears limited by vegetation structure and the availability of locations to anchor webs (*Rypstra, 1983*; *Miyashita et al., 2004*; *Takada et al., 2008*). In the absence of deer, anchoring locations are abundant. In these circumstances, web-building spider abundance will often be limited by prey availability (*Rypstra, 1983*; *Wise, 1993*). The impact of deer on vegetation, spiders, and prey availability is likely sequential. Deer overabundance and subsequent herbivory decreases vegetation complexity. Deer, therefore, lower the availability of web-anchoring locations for spiders reducing spider abundance and predation. This, in turn, increases prey availability in areas with high deer abundance. Following this logic, if spiders were selecting habitat based on prey availability, higher spider abundance would be expected in areas with deer present.

There is a plausible alternative explanation for the increase in prey density. It could merely reflect additional space use by prey in the absence of structural impediments created by vegetation. By this explanation, deer reduce vegetation structure, allowing for more invertebrates to fly unimpeded through plots. This results in larger numbers of prey being captured by sticky traps. Reduced capture and predation of prey by spiders does
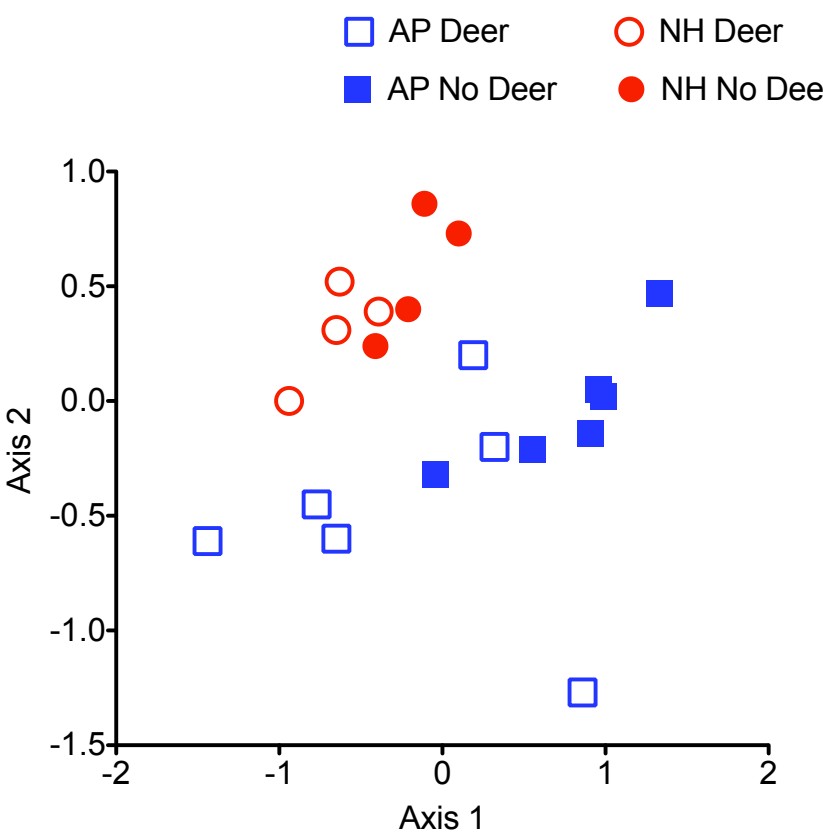

**Figure 5** **NMDS ordination of each web type found in control (open) and exclosure (solid) plots in the Allegheny Plateau (AP) and Northern Highlands (NH) region.** Only the two most variable dimensions of a 3-dimensional solution are shown.

not need to be invoked to account for the increase in invertebrate prey. To determine the relative contributions of spider predation and vegetation structure on prey availability, a future study should manipulate both factors in a 2 × 2 factorial experiment.

A visual inspection of sample plots in spider assemblage space (Fig. 5) suggests modest effects of deer on community structure, despite statistical significance. The absence of strong clustering of sample plots into those with and without deer probably reflects the effects of deer on nearly all web structures (Fig. 4). In plots with deer, the decline in spider web abundance is more or less evenly spread across all web types. As a result, community structure remains little changed as a more or less random loss of individual webs as the capacity of the habitat to support web-building spiders that rely on complex vegetation structure declines. Only the small, dense webs of funnel-weavers were unaffected. Funnel weavers tend to construct their webs at ground level, so these spiders are the least likely to be affected by changes in vegetation architecture.

Other researchers report that the abundance and/or richness of web-building spiders increased in forest areas where deer were experimentally excluded (*Miyashita et al., 2004*; *Suominen & Danell, 2006*; *Takada et al., 2008*). Under the closed forest canopies of our study region, the density of shrubs and tree saplings is often low, owing to low light levels, recalcitrant understory layers that inhibit woody plant growth, and feeding by deer

(*Horsley, Stout & DeCalesta, 2003*; *Sage, Porter & Underwood, 2003*; *Royo & Carson, 2006*). Hence, there are few opportunities for web-building spiders to anchor their webs. Recruitment of web-building spiders might therefore become increasingly disturbance-dependent because large canopy disturbances increase the abundance and density of woody vegetation in the understory. These episodic disturbances could provide key web building substrates lacking in intact forest in areas where deer are abundant. While episodic recruitment is a common feature of marine systems and disturbance-dependent plant populations, it is possible that the widespread overabundance of deer have created conditions in which terrestrial predatory arthropods may develop this same recruitment pattern.

In their review of the effects of deer on ecosystems, *Rooney & Waller (2003)* differentiated between indirect effects due to modified food web interactions, and indirect effects arising from habitat modification. In this study, we can attribute changes in the distribution and abundance of web-building spiders (except funnel web builders) to habitat modification. We believe causality may occur via multiple pathways. Reductions in the density of spider webs probably contributed at least in part to the rise in arthropod prey we observed. In other words, habitat modification can lead to modified food web interactions. The distinction between habitat modification and modified food web interactions should not be viewed as a strict dichotomy. The linkages from deer to vegetation are trophic; deer reduce the density of saplings (*Begley-Miller et al., 2014*) that provide web anchoring points. This structural change altered the abundance of web-building spiders at the third or fourth trophic level (depending on prey), which may have increased the abundance of arthropod prey at the second or third trophic level. *Nuttle et al. (2011)* introduced the concept of a trophic ricochet to describe a top-down effect that altered plant communities, but did not terminate at the lowest trophic level. Instead, it was transformed to a bottom-up effect that persisted for several decades. In this study, we observe a different type of trophic ricochet: a top-down effect of deer on vegetation structure that indirectly affected organisms at multiple trophic levels. Deer directly reduced habitat quality and indirectly reduced the abundance of predators, albeit arthropod predators that do not feed on deer, which in turn apparently resulted in an increase in arthropod prey. Trophic ricochets may be a widespread response when dominant species, keystone species, or ecosystem engineers modify the habitats.

## CONCLUSION

Deer browsing has profound implications for web-building spiders in the forest understory layer. Of the families of spiders we studied, only the funnel web-builders appeared unaffected. Deer greatly modified habitat structure, reducing opportunities for spiders to anchor webs. This reduced the density of spider webs, and in turn led to a 2.7-fold increase in spider prey abundance. Thus, deer herbivory indirectly altered arthropod predator–prey interactions throughout the forest understory. These changes are probably not unique to our study sites, but instead reflect changes throughout both the Allegheny Plateau and Northern Highland regions. Deer populations have increased in both regions

since the 1970s (*Ripple, Rooney & Beschta, 2010*). The effect sizes we observed in our study are perhaps larger than occur throughout the region, because exclosures create the artificial condition of vegetation development in the absence of deer. However, we do not know if web-building spider abundance responds to thresholds in deer abundance, or whether such responses are linear. Studies that take advantage of natural gradients in deer abundance might provide better estimates of contemporary effect sizes, although such correlative studies have limitations.

## ACKNOWLEDGEMENTS

We thank John Wenzel, John Dzemyan, Rhett Rautsaw, Maureen McGeean Lake, and Tim Hanson for helping make this project possible. We also thank Dairymen's Inc. for access to their property. Volker Bahn performed the dispersion analysis in R. Chris Habeck, David Wise, Michael S. Singer, and an anonymous reviewer provided valuable suggestions for improving an early draft of this manuscript.

### Funding

Funding was provided in part by Powdermill Nature Reserve through an internship/fellowship to MJ Chips, and in part by a consulting contract from Dairymen's Inc. to TP Rooney. The funders had no role in study design, data collection and analysis, decision to publish, or preparation of the manuscript.

### Grant Disclosures

The following grant information was disclosed by the authors:
Powdermill Nature Reserve.
Dairymen's Inc.

### Competing Interests

The authors declare there are no competing interests.

### Author Contributions

- Elizabeth J. Roberson and Michael J. Chips conceived and designed the experiments, performed the experiments, analyzed the data, wrote the paper, reviewed drafts of the paper.
- Walter P. Carson conceived and designed the experiments, wrote the paper, reviewed drafts of the paper.
- Thomas P. Rooney analyzed the data, wrote the paper, prepared figures and/or tables, reviewed drafts of the paper.

### Data Availability

   Raw data are deposited at Figshare. DOI: 10.6084/m9.figshare.3438344.

The spreadsheet used to calculate results presented in Fig. 4 is deposited at Figshare. DOI: 10.6084/m9.figshare.3767394.

R code used is deposited at Figshare. DOI: 10.6084/m9.figshare.3804102.

## Supplemental Information

Supplemental information for this article can be found online at http://dx.doi.org/10.7717/peerj.2538#supplemental-information.

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
