# Peer review of "Deer herbivory reduces web-building spider abundance by simplifying forest vegetation structure"

_PeerJ, doi:10.7717/peerj.2538_

## Round 0.1 · original submission · Major Revisions

· Academic Editor

Major Revisions

All three reviewers and I agree that this manuscript reports a nice piece of work that makes a valuable contribution to its field. Although I am asking for "major revisions," most of the revisions are relatively minor. Please follow the suggestions of the reviewers as closely as possible in making your revisions. Reviewers 1 and 2, in particular, have given much constructively critical guidance on how to make these revisions (please see their annotated versions of the original manuscript). Perhaps most importantly, I agree with both of them that the statistical analyses need some revision (e.g., PERMANOVA rather than ANOSIM, and some additional analyses that they specifically ask for). If you see fault with their suggestions, please justify your decision not to follow them. In addition, reviewer 2 caught a major error in the Abstract that needs attention. I agree with all of the reviewers' comments on copy editing issues, and will add one more: check the spelling of authors' names cited. For example, Wootton is written as Wooten, and Habeck is written as Habek. Finally, I think reviewer 2 gives some excellent editing suggestions in the annotated version of your manuscript. I encourage you to follow those recommendations to improve the writing style and clarity.

·

Basic reporting

The authors report on the direct and indirect effects of deer on spider communities in two forest ecosystems in North America. They successfully show that deer reduce web density with an observed cascading effect on spider prey abundance. The writing is generally clear save for some organizational issues, and a few minor grammatical and omission errors. The Introduction provides an appropriate set-up for the theme of their work. Some minor suggestions are included as mark-ups in the pdf. Mean ans SE for data in Figure 2 should be included somewhere in the manuscript. Figure 3 should have R2, p and regression equation added (or provide these in the Results section).

Experimental design

The experimental design is robust. However, I have included a few suggestions to the authors regarding additional analyses. For instance, I believe the manuscript would be improved if they re-ran some of their analyses without funnel web data, and if they switched from ANOSIM to PERMANOVA. In addition, the description of the Methods is lacking and/or inaccurate in places (e.g., meta-analytical modelling procedures). These issues are easily surmountable and do not detract from the overall quality of the experiment.

Validity of the findings

No major issues here. Suggestions are included as mark-up in the attached pdf.

Additional comments

I really do believe that more analyses would improve this ms, which is why I am suggesting major revisions. Firstly, the funnel spiders are obviously not linked to WSAI at 0.5 and 1m. Including them in further analysis without some ground-level variable describing disturbance seems useless to me, and may even influence inference about the other web guilds. Also, the benefits of PERMANOVA and Dispersion analysis seem a better fit for your multivariate analysis than ANOSIM, particularly given that you suspect dispersion differences in your regions. Otherwise, this is a well written and interesting manuscript with valuable information on the indirect effects of deer overabundance. Congrats!

·

Basic reporting

Raw data need to be made available.

Experimental design

OK

Validity of the findings

See comments below -- inconsistency between Abstract and body of the manuscript with respect to a major finding of the study.

Additional comments

This is an excellent study and a solid manuscript, though there are some revisions that are required to adequately highlight its novel findings.

The major change is to correct the incorrect statement in the Abstract about the effect of the deer exclosures on prey abundance (the actual result is opposite to what is in the Abstract, according to the Results and Discussion sections), and rearrange the Discussion to emphasize the finding that deer indirectly reduced spiders which then likely led to an increase in prey. This is an exciting finding and does not appear until the end of the manuscript!! It should be emphasized in the Abstract, probably in the Title, and made the main focus of the Discussion.

I would also reconsider the discussion of the multivariate patterns --- I like the conservative approach, but would also consider also conducting a PCo and PERMANOVA – I wouldn’t restrict the analysis to ANOVSIM. It’s not clear why Bray Curtis or sq root transformation was used – are the results similar with other data transformations?

I’m also not sure why the meta-analysis technique was used – seems too complex for the data.

I may have missed it, but there is no reference to where the data are archived. I believe this is a requirement of PeerJ.

I’ve made comments on the attached PDF.

Reviewer 3 ·

Basic reporting

See below.

Experimental design

See below.

Validity of the findings

See below.

Additional comments

I found this an interesting and stimulating paper. The sampling is well designed and the results are appropriately analysed. I am glad to recommend publication after the following issues have been addressed (numbers refer to page/line numbers):
6/14: Insert ‘the’
7/16-20: You need to explain how your study is different to, and adds useful insight to, Miyashita et al and Takada et al. The titles seem to be somewhat similar.
9/23: Insert ‘we’
10/7: The method is unclear. Is the stick rotated horizontally?
10/18 What was measured here? Total body length, of each specimen?
11/2: Why divide by 7.5? Would it not be better to make ‘days’ an extra factor in your analysis?
11/10 You say ‘improve normality’ – so were the data adequately normalised?
11/16 Insert ‘of’
12/1: Delete ‘used’
13/17: You haven’t really explained why there would be more prey in the deer plots. Could it be simply that the prey (mostly insects?) were freer to move around because the foliage was less dense and so they were more likely to get caught in the traps?
14/4: You refer to a 3D solution but this needs explaining.
15/2: Replace ‘therefore’ with ‘in these circumstances’ (i.e. in absence of deer)
15/10 ‘declines’
16/20: The idea that a reduction in spiders has caused an increase in prey is only speculation; you have not tested this experimentally. It could be the results of completely independent factors driving these effects.

---

## Round 0.2 · Minor Revisions

· Academic Editor

Minor Revisions

Congratulations on producing a nicely revised manuscript in accordance with the extensive comments of the reviewers! I am pleased to see your many thoughtful responses to the reviewers' comments (including cases in which you did not follow the specific recommendation of a particular reviewer). I like your approach to dealing with contradictory comments by different reviewers; presenting alternative explanations is the most fair-minded choice.

The extra figures generated for the cover letter would be good to include as supplemental figures in an appendix.

I am confident that this manuscript will make an important contribution to the growing body of literature on ecological effects of deer overbrowsing. However, there are still several minor changes/corrections that you need to make before final acceptance.

These include the following:

1. Use the correct spelling of Wootton in the Wootton 1994 reference throughout (lines 42, 49, bibliography).
2. Line 46, change "are" to "were."
3. Line 62, comma after "Similarly."
4. Line 63, extra space after "mice."
5. Line 100, change to "1.8-m tall."
6. Lines 109-110, change to "2.25-m tall."
7. Line 117 and legend for figure 1, insert comma after "(Uloboridae)."
8. Line 131, change the comma to a semicolon after "location." Alternatively, keep the comma and insert "and."
9. Line 133, delete "or not" after "whether."
10. Line 147, change to "1-m high."
11. Line 183, change "their" to "its."
12. Line 194, change "7 web type abundance" to something clearer and easier to read.
13. Line 203, extra space after "(Clarke and Gorley 2006)."
14. Line 253, change "have" to "had."
15. Lines 269-270, change to: "...were selecting habitat based on prey availability, higher spider abundance would be expected in areas with deer present."
16. Line 289, change "are" to "were."
17. Line 303, change "form" to "from."
18. Lines 304-305, the sentence "We believe this is not simply cause and effect" is problematic because it doesn't accurately convey the meaning that I believe you intend. I believe you mean to say something like "We believe causality may occur via multiple pathways." I'm not wedded to my suggestion, but the sentence needs to be changed because you are, in fact, arguing that there is a causal relationship; the point is that it may be more complicated than either one or the other.
19. Line 315, add "a" after "observe."
20. Figure 4 legend, the second sentence is redundant with the information in the figure. I suggest removing it. Also, change "p" to "P" in reporting the P-value of your test.
21. Figure 5 legend, the first sentence is unclear in relation to the figure. I suggest something like "NMDS ordination of the abundance of each spider web type found in..."

---

## Round 0.3 · accepted · Accept

· Academic Editor

Accept

I'm pleased to accept this manuscript, and look forward to its worthy contribution to the field. Great work!